# Effects of Eggshell Membrane on Keratinocyte Differentiation and Skin Aging In Vitro and In Vivo

**DOI:** 10.3390/nu13072144

**Published:** 2021-06-22

**Authors:** Kyohei Furukawa, Masaya Kono, Tetsuro Kataoka, Yukio Hasebe, Huijuan Jia, Hisanori Kato

**Affiliations:** 1Health Nutrition, Department of Applied Biological Chemistry, Graduate School of Agricultural and Life Sciences, The University of Tokyo, 1-1-1 Yayoi, Bunkyo-ku, Tokyo 113-8657, Japan; k-furukawa@g.ecc.u-tokyo.ac.jp (K.F.); msy.k.1203@gmail.com (M.K.); tetu19981119@gmail.com (T.K.); 2ALMADO Inc., Tokyo 104-0031, Japan; yukio@almado.co.jp

**Keywords:** eggshell membrane, keratinocyte differentiation, TRPV, skin aging, skin thickness

## Abstract

Skin aging is one of the hallmarks of the aging process that causes physiological and morphological changes. Recently, several nutritional studies were conducted to delay or suppress the aging process. This study investigated whether nutritional supplementation of the eggshell membrane (ESM) has a beneficial effect on maintaining skin health and improving the skin aging process in vitro using neonatal normal human epidermal keratinocytes (NHEK-Neo) and in vivo using interleukin-10 knockout (IL-10 KO) mice. In NHEK-Neo cells, 1 mg/mL of enzymatically hydrolyzed ESM (eESM) upregulated the expression of keratinocyte differentiation markers, including keratin 1, filaggrin and involucrin, and changed the keratinocyte morphology. In IL-10 KO mice, oral supplementation of 8% powdered-ESM (pESM) upregulated the expression of growth factors, including transforming growth factor β1, platelet-derived growth factor-β and connective tissue growth factor, and suppressed skin thinning. Furthermore, voltage-gated calcium channel, transient receptor potential cation channel subfamily V members were upregulated by eESM treatment in NHEK-Neo cells and pESM supplementation in IL-10 KO mice. Collectively, these data suggest that ESM has an important role in improving skin health and aging, possibly via upregulating calcium signaling.

## 1. Introduction

The aging population is increasing rapidly, especially in developed countries, and the global population aged over 60 is expected to increase by 22% in 2050 [1]. Aging affects cellular metabolism, physiology and function [2], increases disease morbidity and mortality [3] and decreases the health-related quality of life (HRQoL). Skin aging, one of the hallmarks of the aging process, leads to skin wrinkling, hair graying, scalp hair thinning/baldness, thin skin, etc. These physiological changes are caused by the accumulation of macromolecular damage, impairment of tissue renewal, loss of physiological function integrity [4]. Skin aging is mainly divided into two categories; chronological aging and photo-aging [5]. Chronological aging is occurred by internal factors (such as hormone levels, genotypes, etc.) and is hard to regulate, as these factors naturally change with age. On the contrary, photo-aging is altered by external factors, such as ultraviolet radiation, nutritional levels, chemical pollution; therefore, it can be delayed by interventions of balanced nutrition, functional bioactive compounds and skincare products [4,6].

Eggshell membrane (ESM) is a natural byproduct of egg processing and is usually discarded as industrial waste, resulting in environmental load. However, as ESM is rich in fibrous protein, collagen, hyaluronic acid, chondroitin sulfate and glucosamine [7], it is possible to utilize ESM as a functional bioactive compound in health science. Previous studies indicated that oral supplementation of ESM is safe and has an anti-inflammatory and anti-obese effect in rodents [8,9,10]. Given that inflammation is closely linked with skin health along with the fact that collagen peptides have a positive effect against skin aging [4], we hypothesized that ESM supplementation could improve skin aging and health.

Therefore, to test this hypothesis, this study investigated the effects of powdered as well as hydrolyzed ESM on skin-derived cells using neonatal normal human epidermal keratinocytes (NHEK-Neo). Then we tested the effect of powdered ESM in interleukin-10 knockout (IL-10 KO) mice. While IL-10 KO mice are used as model for inflammatory bowel disease, it is considered as one of the frail and fragility model since it developed osteopenia and hair loss, reduces bone formation and energy metabolism, and increase levels of several inflammatory cytokines [11,12,13,14]. We explored the role and possible mechanism of ESM in keratinocyte differentiation and improved skin health and aging, which could provide an effective and functional source of bioactive compounds to improve the HRQoL of older people.

## 2. Materials and Methods

### 2.1. Cell Culture

Neonatal Normal Human Epidermal Keratinocytes (NHEK-Neo) were purchased from LONZA Japan (Tokyo, Japan). The cells were cultured at 37 °C under 95% air/5% CO_2_ with KGM™ Gold Keratinocyte Growth Medium BulletKit™ (#00192060, LONZA), containing KBM™ Gold™ Basal Medium and KGM™ Gold™ SingleQuots™ supplements. The medium was replaced every 2 days during the experimental periods.

### 2.2. Determination of Cell Cytotoxicity

Cell cytotoxicity was determined using the lactate dehydrogenase (LDH) assay (#299-50601, Wako Pure Chemical Industries, Osaka, Japan). The cells were seeded onto Biocoat™ Collagen I 24-well plate (Corning Inc., Corning, NY, USA) at 2.5 × 10^4^ cells/well and cultured for 2 days with KGM™ Gold Medium. The medium was replaced with KGM™ Gold Medium containing either powdered ESM (pESM; 0.1, 0.5 or 1 mg/mL), alkaline (sodium hydroxide)-hydrolyzed ESM (hESM; 1, 2 or 3 mg/mL), enzymatically-hydrolyzed ESM (eESM; 1, 2 or 3 mg/mL) or 0.2% Tween-20 (positive control). In the preparation of eESM, pESM was treated with 5.5 mg/g sodium metabisulfite for 8–10 h and heated at 90 °C for 6 h, then hydrolyzed with 3.6 mg/g papain at 60 °C for 3 h. The three types of ESMs were obtained from ALMADO Inc. (Tokyo, Japan). The amino acid compositions of each ESM are shown in Appendix A. The plates were then incubated at 37 °C for 24 h, the medium was collected into 1.5 mL tubes and cellular LDH was also collected with the medium containing 1% Tween-20. The five times diluted samples were used for the LDH assay. The assay was performed according to the manufacturer’s instructions, and the absorbance (Ab) was measured at 560 nm using a spectrophotometer (Thermo Scientific, Waltham, MA, USA). The cytotoxicity was calculated using the following formula.
Cytotoxicity (%) = (Abs in medium)/(Abs in medium + Abs in cells) × 100

### 2.3. Morphological Changes

The cells were seeded onto Biocoat™ Collagen I 100-mm dish (Corning Inc., Corning, NY, USA) at 2.0 × 10^5^ cells/dish and cultured for 1 day. The medium was replaced with pESM, hESM or eESM containing medium at 2 days’ interval and cultured for 4 days. The morphological changes were recorded by monitoring the cells by a microscope (Leica Microsystems GmbH, Wetzlar, Germany).

### 2.4. Animal Experiment

Male B6.129P2-IL-10<tm1Cgn>/J mice and female C57BL/6J mice were purchased from The Jackson Laboratory and Charles River Japan, respectively. Female IL-10-Hetero KO mice (F1) obtained by mating the two aforementioned mice were further mated with male IL-10-Homo KO mice (F1). The male IL-10-Homo KO mice (F2, KO) were used in this study. The mice were individually housed in cages in a temperature-controlled (23 ± 2 °C), 12 h light-dark cycle environment. Four-week-old KO and C57BL/6J (WT) mice were fed an AIN-93-based diet or an 8% pESM-supplemented diet (KOE) for 28 weeks. The diet composition is shown in Table 1, which is the same as that reported in a previous study [8]. After the experimental period, the back skin was collected under deep anesthesia with isoflurane, removed the dermal white adipose tissue from collected skin sample, and stored at −80 °C until further analysis.

### 2.5. Morphological Analysis

Hematoxylin and eosin (H&E) staining and Fontana-Masson staining were carried out to evaluate skin thickness and melanin content in the back skin samples of mice, respectively. For H&E staining, the optimal cutting temperature (OCT) compound (Sakura Finetek, Torrance, CA, USA) embedded 5-µm-thick sliced sections were used. The stained sections were then decolorized with ethanol and enclosed using Mount-Quick (Daido Sangyo, Tokyo, Japan). Tissue images were obtained using light microscopy (Olympus BX51 microscope, Olympus Optical, Tokyo, Japan), and the thickness (total of epidermis and dermis) was measured randomly at six different parts of each skin sample.

For Fontana-Masson Staining, the frozen sections were reacted with Fontana ammonia silver aqueous under shading conditions overnight at room temperature. After washing using distilled water, the sections were reacted with 0.2% gold chloride aqueous and fixing solution for 5 s and 1 min, respectively. Subsequently, the sections were washed with distilled water and reacted with Kernechtrot solution to stain nuclei. After washing, the sections were enclosed, and tissue images were obtained as described for the H&E staining.

### 2.6. RNA Extraction from Cell and Tissue Samples

RNA extraction from cell sample was performed using Nucleospin^®^ RNA plus kit (Takara Bio Inc., Shiga, Japan). The cells were seeded onto a Biocoat™ Collagen I 6-well plate (Corning Inc., Corning, NY, USA) and cultured for 1 day. Then, the medium was replaced with different types of ESM-containing media and cultured until 80–90% confluence.

RNA extraction from animal samples was carried out with Nucleospin^®^ RNA (Takara Bio Inc., Shiga, Japan). In the animal experiment, the frozen skin sample was crushed with a cool mill (AS ONE, Osaka, Japan), which was cooled in liquid nitrogen before use. Crushed samples were homogenized with 1 mL of TRIzol reagent, and then 200 µL of chloroform was added. The mixture was centrifuged at 12,000× *g*, 4 °C for 15 min, and the supernatant was mixed with Nucleospin^®^ RNA RA1 buffer and ethanol. Further experimental procedures were per the manufacturer’s instructions.

Total RNA content was measured with NanoDrop ND-1000 (Thermo Scientific, Waltham, MA, USA). For DNA microarray analysis, RNA quality was checked by Agilent 2100 Bioanalyzer (Agilent Technologies, Santa Clara, CA, USA), and all samples were confirmed to have an RNA integrity number of more than 6.0.

### 2.7. DNA Microarray Analysis

In total, 100 ng pooled RNA isolated from NHEK-Neo and 250 ng of pooled RNA isolated from mouse skin samples were used for microarray analysis with Clariom™ S Array, human and Affymetrix Genechip™ Mouse Genome 430 2.0 Array(Applied Biosystems™, Waltham, MA, USA), respectively. RNA samples derived from cells and mice were reverse-transcribed to single-strand cDNA using Genechip™ WT Amplification Kit (Affymetrix) and Genechip™ 3′ IVT Express Kit (Thermo Scientific, Waltham, MA, USA), respectively. The synthesis of double-strand cDNA and cRNA was performed according to the manufacturer’s instructions. The cRNA samples were hybridized onto the array chip at 45 °C for 16 h and stained with streptavidin-phycoerythrin using GeneChip™ Fluidics Station 450 (Applied Biosystems™). The image data was detected with GeneChip™ Scanner 3000 (Affymetrix).

The data were converted to CEL file format using Affymetrix GeneChip^®^ Command Console. In the cell experiment, the data were normalized with Transcriptome Analysis Console. The numerical conversion and the normalization of the animal experiment data were performed using R ver. 3.5.1 and Affymetrix Microarray Analysis Suite 5 (Affymetrix), respectively. The cut-off condition was set up as |Fold change| ≥ 2 in the cell experiment, and |Fold change| ≥ 1.5 and signal intensity ≥ 40 in the animal experiment. The gene sets satisfying these cut-off conditions were further subject to an ingenuity pathway analysis software (IPA, http://www.ingenuity.com/, date accessed: 29 September 2020) for the bioinformatics analysis including canonical pathway analysis, disease and function analysis and upstream analysis.

### 2.8. Real-Time Reverse Transcription-Polymerase Chain Reaction (Real Time RT-PCR)

To validate the microarray data, the expression of some selected genes was measured using real-time RT-PCR. RNA samples derived from the cells and mice were reverse-transcribed to cDNA using Primescript™ RT Master Mix (Takara Bio Inc.), according to the manufacturer’s instructions. The cDNA was then mixed with appropriate primers and SYBR^®^ Premix Ex Taq™ (Takara Bio Inc.), and real time RT-PCR was performed using the Thermal Cycler Dice Real Time System TP800 (Takara Bio Inc.). The primer sequences are shown in Appendix A. The obtained data were normalized with B2M (Human Housekeeping Gene Primer Set, #3790, Takara Bio Inc.) and *Gapdh* in the cell and animal experiments, respectively. The results are expressed as fold-change values compared with the control or WT group.

### 2.9. Statistical Analysis

Results are expressed as the mean ± standard error. All data were analyzed with one-way analysis of variance (ANOVA), followed by a Tukey’s test, and *p*-value < 0.05 was considered to indicate statistical significance.

## 3. Results

### 3.1. Effects of Different Types of ESM on Cytotoxicity and Cell Morphology

As pESM contains 91~94% indigestible protein and approximately 46% of them are digested and absorbed in the digestive tract [8], we tested the effects of three types of ESM, pESM, hESM and eESM, on cytotoxicity and cell morphology in cell culture experiments. As shown in Figure 1a, the cytotoxicity based on LDH assay was not increased by three types of ESM in the tested concentrations. However, the value was significantly decreased by 1 mg/mL of pESM, 2 and 3 mg/mL of hESM, and 1–3 mg/mL of eESM in NHEK-neo cells. Therefore, we selected 1 mg/mL supplementation for subsequent experiments in the hESM and eESM groups. In the pESM group, we chose 0.1 mg/mL supplementation, as the insoluble materials disturbed the cell observation when the medium was supplemented with 1 mg/mL pESM.

Figure 1b shows the time course of morphological changes in ESMs supplemented NHEK-Neo cells. pESM and hESM altered the cell morphology neither on Day 1 nor 4 compared with the control group. Notably, eESM-treated cells showed the flattening-like changes on Day 4 of treatment. These results suggested eESM might promote cell differentiation in NHEK-Neo cells.

### 3.2. Microarray Analysis in Cell Culture Experiments

We carried out DNA microarray analysis to further investigate the mechanism underlying eESM-induced cell differentiation in NHEK-Neo cells. As shown in Figure 2a, the expression of many genes was altered by eESM supplementation (1289 increased and 1330 decreased) compared with the control group. To clarify the positive effect of eESM, we uploaded the altered genes to IPA. Previous studies indicated that keratinocyte differentiation is accompanied by the decreases in the synthesis of nucleotide and protein [15,16]. A similar trend was also observed in our microarray analysis. As illustrated in Figure 2b, the canonical pathway analysis indicated that the pathway related to cell cycle and nucleotide biosynthesis such as “Cell Cycle Control of Chromosomal Replication”, “Purine Nucleotide De Novo Biosynthesis II” and “Cyclins and Cell Cycle Regulation” were downregulated by eESM.

Furthermore, disease and function analysis identified that “Differentiation of Skin” and “Formation of Skin” were upregulated (Figure 2c). Since both of these pathways included similar gene sets, we focused on “Formation of Skin,” which contained many gene sets. The major gene list is shown in Table 2. Notably, eESM upregulated the differentiation markers of keratinocytes such as involucrin (*IVL*), filaggrin (*FLG*) and keratin 1 (*KRT1*). Then, we carried out real time RT-PCR analysis to further evaluate the differentiation markers in ESM-treated keratinocytes. As shown in Figure 2d, gene markers of the basal layer, *KRT5* and *KRT14,* were not altered by any ESM supplementation. However, levels of markers of the spinous layer, *KRT1*, *KRT10* and *IVL*, were significantly increased by eESM supplementation compared to the control group, consistent with the results of microarray analysis. Additionally, hESM significantly increased the expression of these genes, while the increments were significantly lower than those in the eESM group. On the contrary, pESM treatment did not show significant changes in the expression of these genes (*p* > 0.10). The expression level of the marker of granular keratinocytes *FLG* was significantly increased by eESM supplementation, consistent with the microarray data. The expression of *FLG* was also significantly increased by pESM but not hESM (*p* > 0.10).

Moreover, in the microarray data, eESM upregulated Kallikrein-related peptidase (*KLK*)*-5* and *-7* and serine protease inhibitor kazal type (*SPINK*)*-5* expression (Table 2), which are inducible in epidermal keratinocytes during cell differentiation. Similar results were observed in real-time RT-PCR analysis, while these increments were not observed in pESM or hESM-treated cells (*p* > 0.10, Figure 2e). These lines of data indicated that eESM supplementation induced cell differentiation.

### 3.3. Possible Involvement of Calcium Signaling in eESM-Induced Keratinocyte Differentiation

Microarray analysis indicated that *DSG1* and *CTSV* were upregulated by eESM supplementation (Table 2), and similar results were obtained by real-time RT-PCR (Figure 2e). These genes are related to keratinocyte differentiation, skin disease and calcium binding. Since the keratinocyte differentiation is triggered by extracellular calcium ion or calcium signaling [15,16], we investigated the gene expression of the transient receptor potential (TRP) channel families TRPV3, TRPV4 and TRPV6. As shown in Figure 2f, *TRPV3* expression was significantly increased by eESM supplementation compared with the control group, while no significant differences were observed in pESM- or hESM-treated cells. *TRPV4* and *TRPV6* levels were not altered by any ESM supplementation.

### 3.4. Effects of pESM Supplementation on Skin Aging in IL-10 KO Mice

We next investigated whether pESM supplementation could ameliorate skin aging in IL-10 KO mice. As shown in Figure 3a,b, skin thickness evaluated by H&E staining was significantly decreased in IL-10 KO mice compared with WT mice. Notably, pESM supplementation significantly improved the value near to that in WT mice. Even though the decline of melanin formation is considered to associate with the pathogenesis of silver hair, Fontana-Masson staining showed no obvious changes in response to IL-10 KO and ESM supplementation (Figure 3a). These results indicated that oral supplementation of ESM ameliorated skin thinning in IL-10 KO mice.

### 3.5. Microarray Analysis in pESM-Supplemented IL-10 KO Mice

We performed the microarray analysis to investigate the molecular mechanism of pESM’s effects in IL-10 KO mice. The data indicated that 2232 and 1567 probe sets were upregulated and downregulated in IL-10 KO mice compared to the WT group, respectively. Here, pESM supplementation upregulated 2067 probe sets and downregulated 195 probe sets. Then, we performed further analysis of the altered genes using IPA software. As shown in Figure 4a, in the comparison between the WT and KO groups, the inflammation-related pathways were upregulated in the KO group, such as “p38 MAPK Signaling”, “Toll-like Receptor Signaling”, “Acute Phase Response Signaling” and “NF-κβ Signaling”. However, in comparison between the KO and KOE groups, these pathways were not listed up. As shown in Figure 4b, pESM supplementation upregulated “Calcium Signaling”, which is consistent with the results of in vitro study. The list of the major genes is shown in Table 3. In this case, pESM supplementation upregulated calcium ion channel-related genes, and *Trpv6* is also listed in the gene list. Real time RT-PCR analysis showed a similar tendency, while the expression levels were not significant (Figure 4d). Other TRPVs, *Trpv3* and *Trpv4*, were also upregulated two-fold compared to the KO group; however, the increments did not reach statistical significance. The expression levels of these three genes were significantly lower in the KO and KOE groups compared with the WT group.

Furthermore, pESM supplementation upregulated “GP6 Signaling” (Figure 4b), which has a crucial role in collagen synthesis. Similar results were observed in Disease and Function analysis. As shown in Figure 4c, pESM supplementation upregulated “Proliferation of connective tissue cells”, “Growth of epithelial tissue” and “Differentiation of connective tissue cells”. The list of major genes related to the “Growth of epithelial tissue” is shown in Appendix A. It contains collagen synthesis-related genes, such as Collagen type I alpha (*Col1a*)*-1* and *-2*, and Collagen type XVIII alpha 1 (*Col18a1*). Fibroblast growth factor 1 (*Fgf1*) was also upregulated by pESM supplementation, which relates to collagen synthesis and skin elasticity [17]. Of these, we measured *Col1a1* expression by real time RT-PCR analysis. As shown in Figure 4e, Col1a1 was increased by pESM supplementation, consistent with the results of microarray analysis, while it did not reach the statistical significance. pESM upregulated Wnt family members (*Wnt*)*-5a* and *-7a* (Appendix A), which have been identified as the skin aging biomarker and suppressive inducer against cell senescence [18,19].

We further examined the expression of skin homeostasis- and aging-related genes. The expression of telomerase reverse transcriptase (*Tert*), which induces proliferation of hair follicle bulge stem cells and hair growth [20], was significantly decreased in the KO group compared with the WT group. In the KOE group, the expression showed the intermediate value, whereas there was no significant difference compared to the KOE and KO groups (*p* > 0.10, Figure 4e). Transforming growth factor (TGF)-β signaling is considered to relate to healthy skin homeostasis, hair growth and collagen synthesis [20,21]. In our data, *Tgfb2* and *Tgfb3* were not altered in all the tested groups, while *Tgfb1* was significantly increased in the KO group, and its increment was tended to decline in the KOE group (*p* = 0.07). In addition to TGF-β1, platelet-derived growth factor (PDGF)-β is crucial in cell function in connective tissues and relates to fibrosis [22]. The expression of *Pdgfb* was significantly increased in the KO group, while the increase was tended to be suppressed in the KOE group (*p* = 0.08). TGF-β also induces connective tissue growth factor (CTGF) expression, which induces organ fibrosis, including skin [23]. Similar results were observed in *Ctgf* expression, which was significantly increased in the KO group, and its increment was declined in the KOE group (*p* = 0.07). Furthermore, based on cellular experiments, we investigated the expression of *Klk5* and *Klk7* in the microarray dataset, while the data showed that the expressions were not increased by pESM supplementation (*Klk5*, fold change (KOE vs. KO): 0.98; *Klk7*, fold change (KOE vs. KO): 0.89).

## 4. Discussion

The present study indicated that eESM induces keratinocyte differentiation in NHEK-Neo cells and that pESM supplementation improves the decline in skin thickness in IL-10 KO. These findings suggest that ESMs may improve skin health and aging. Previous studies indicated that polyphenols, vitamins and collagen peptides have a beneficial role in skin health, while these effects are mainly observed via anti-oxidative stress and anti-inflammation effects [4]. In contrast, our study shows the direct beneficial effect of ESM on skin health.

Our microarray analysis indicated that calcium signaling might associate with the positive effects of eESM and pESM in cell and animal experiments, respectively. Especially, TRPVs were upregulated in both experiments. Previous studies indicated that calcium or calcium ions are crucial to induce keratinocyte differentiation [15,16]. Furthermore, it has been reported that calcium concentration gradient is established in the epidermis, and it is broken down with age in mice and humans [24,25]. Therefore, the upregulation of calcium signaling could be an important factor to explain the effect of eESM and pESM in our experiment. The calcium concentrations in three types of ESM-supplemented medium using AmpliteTM Fluorimetric Calcium Quantitation Kit Red Fluorescence (AAT Bioquest^®^), did not vary compared with the control group (Control: 0.10 mM, pESM: 0.08 mM, hESM: 0.13 mM, and eESM: 0.23 mM) and much lower than 1.2 mM that induces keratinocyte differentiation in murine and human cell lines [26,27]. Based on these lines of information, it is suggested that at least in NHEK-Neo cells, eESM induces keratinocyte differentiation, possibly via the upregulation of calcium signaling, which might be caused by the transcription of TRPVs rather than extracellular calcium concentrations.

Notably, eESM upregulated calcium signaling and induced keratinocyte differentiation, while this effect was not observed with the same concentration of hESM. The differences could be explained by amino acids or peptides of ESM. As shown in Appendix A, total amino acid concentrations of hESM and eESM were similar, but the concentrations of asparagine, histidine, threonine, citrulline, arginine, tyrosine, phenylalanine and branched-chain amino acids were higher in eESM than in hESM. Even though a previous study has shown that several nutrients and compounds have beneficial roles in skin health [28], there is little information on the relationship among amino acids, calcium signaling and keratinocyte differentiation. Moreover, functional peptides derived from natural resources have beneficial roles in regulating metabolic disorders incurred by aging [29]. Since enzymatic hydrolysis is relatively mild rather than acid hydrolysis, eESM could contain the functional peptides. This study, for the first time, suggests that the amino acids or functional peptides in eESM might associate with the upregulation of calcium signaling and keratinocyte differentiation. Nevertheless, the related mechanisms need to be investigated further.

In the animal experiment, we could not conclude the central mechanism of pESM’s effect based on the DNA microarray data. However, our results imply the following possibilities. First, based on microarray data, genes related to calcium signaling and collagen synthesis might be partly involved in pESM’s effect in the back skin of IL-10 KO mice. Second, based on the *Tert* result, pESM might improve telomere shorting, a hallmark of aging [30]. Third, based on the results of growth factors such as *Tgfb1*, *Pdgfb* and *Ctgf*, pESM might affect fibrosis or inflammation-related genes. However, further investigations are required to clarify the mechanism of pESM’s action using other aging or frailty animal models.

Our previous study indicated that oral supplementation of pESM has several beneficial effects against intestinal inflammation and obesity [8,9]. Taken together, it can be inferred that ESM, a rich source of beneficial bioactive compounds, could improve several diseases and the aging process of the skin. Furthermore, since we indicated that ESMs did not show cell cytotoxicity in at least keratinocytes, it could be proposed that the direct application of eESM to the skin, such as skin cream, could show much impactive effect for skin health and skin aging. Since ESM is regarded as industrial waste and discarded after egg processing, it would be of significance if further in-depth studies could be carried out to explore the effective and functional usage of ESM in our life and health. Furthermore, our study could contribute for a novel cosmeceutical concept “drug repositioning” [31]. Accumulating nutritional study against skin health and disease are still important to prevent skin disorders.

## 5. Conclusions

This study indicates that ESM plays an important role in improving skin health and aging in NHEK-Neo cells and IL-10 KO mice. These findings could contribute to better understand the role of the beneficial compounds for improving skin health, increase HRQoL for elderly people and implement a sustainable food system.

## Figures and Tables

**Figure 1 nutrients-13-02144-f001:**
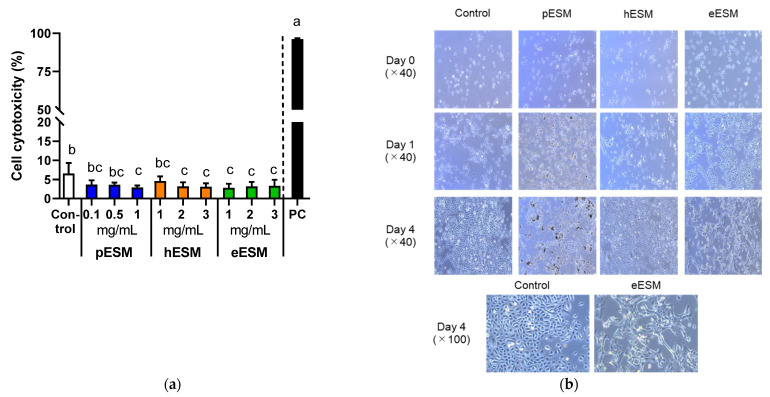
Effects of three types of ESMs on cell cytotoxicity and morphological characterization of NHEK-Neo cells. (**a**) Cell cytotoxicity based on LDH assay in response to different concentrations of pESM, hESM and eESM. 0.2% Tween-20 (*v/v*) was used as the positive control (PC). Data are means ± SD and expressed as % (*n* = 4). The calculation is shown in the Materials and Methods; (**b**) Morphological changes of NHEK-Neo cells in response to 0.1 mg/mL of pESM, and 1 mg/mL of hESM and eESM each. The different lowercase alphabets indicate significant differences between the tested-group (*p* < 0.05).

**Figure 2 nutrients-13-02144-f002:**
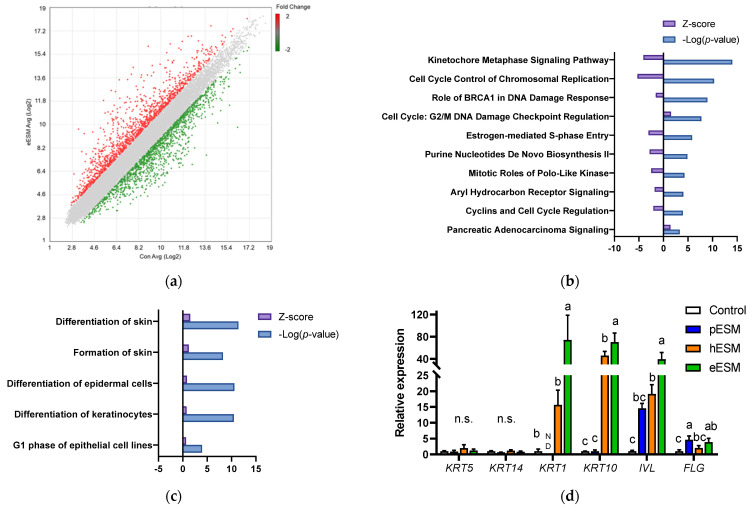
DNA microarray analysis of eESM­treated NHEK­Neo cells. (**a**) Scatter plot of microarray data. Each dot represents the fold changes in gene expression. The values on the X and Y axes are the signal values (Log2 scaled) of the control and eESM groups, respectively; (**b**,**c**) Canonical analysis (**b**) and disease and function analyses (**c**) using IPA software. In both analyses, the cut-off condition was |Activation Z-score| > 1, and data was sorted by −Log (*p*-value); (**d**–**f**) results of real-time RT-PCR on keratinocyte differentiation markers (**d**), genes related to keratinocyte differentiation and calcium signal (**e**) and calcium channel (**f**). Data are means ± SD (*n* = 3) and expressed as fold change values compared with the control group. ND means the expression of the genes was not detectable in our method. The different lowercase alphabets indicate significant differences between the tested-group (*p* < 0.05).; n.s.: not significant.

**Figure 3 nutrients-13-02144-f003:**
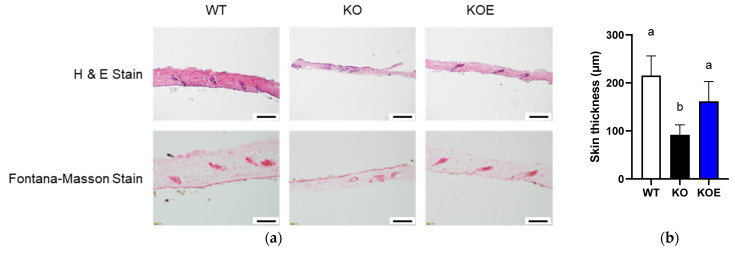
Effects of 8% pESM supplementation on skin morphology in IL-10 KO mice. (**a**) Results of H&E stain (upper side) and Fontana-Masson Stain (lower side); (**b**) Skin thickness based on H&E stain. Data are means ± SD, *n* = 5, 7 and 6 for WT, KO and KOE groups, respectively. The different lowercase alphabets indicate significant differences between the tested-group (*p* < 0.05). WT: Wild type, KO: IL-10 knockout mice, KOE: KO mice supplemented with 8% pESM.

**Figure 4 nutrients-13-02144-f004:**
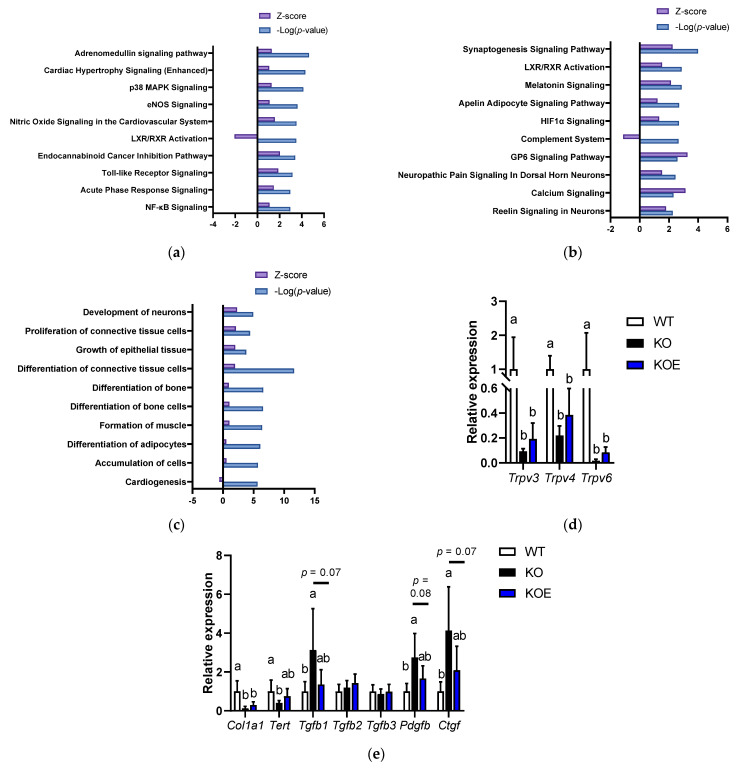
DNA microarray analysis on pESM supplemented IL­10 KO mice. (**a**,**b**) Canonical analysis comparison between the KO and WT groups (**a**) and between the KOE and KO groups (**b**) using IPA software. In both analyses, the cut-off condition was |Activation Z­score| > 1, and data was sorted by −Log (*p*-value); (**c**) Disease and function analysis comparison between the KOE and KO groups using IPA software. The cut-off condition was |Activation Z­score| > 1, and data was sorted by −Log (*p*-value); (**d**,**e**) Results of real-time RT-PCR on calcium channel (**d**) and skin aging-related genes (**e**). Data are means ± SD, *n* = 5, 7 and 6 for WT, KO and KOE groups, respectively, and expressed as fold change values compared with the WT group. WT: Wild type, KO: IL­10 knockout mice, KOE: KO mice supplemented with pESM. The different lowercase alphabets indicate significant differences between the tested-group (*p* < 0.05).

**Table 1 nutrients-13-02144-t001:** Diet composition.

Component (%)	Control Diet	8% pESM Supplemented Diet
Casein	20.0	16.3
β-corn starch	39.7	35.4
α-corn starch	13.2	13.2
Soybean oil	7.0	7.0
Sucrose	10.3	10.3
Cellulose	5.0	5.0
Vitamin mixture *	1.0	1.0
Mineral mixture *	3.5	3.5
L-cystine	0.3	0.3
ESM powder	0.0	8.0

* AIN-93 prescription (Oriental Yeast Co., Ltd., Tokyo, Japan). ESM: Eggshell membrane.

**Table 2 nutrients-13-02144-t002:** List of major genes related to “Formation of skin”.

Gene Symbol	Gene Name	Fold Change (eESM vs. Control)
*ALOX15B*	Arachidonate 15-lipoxygenase type b	2.01
*CDH1*	Cadherin 1	2.17
*CDSN*	Corneodesmosin	48.29
*CST6*	Cystatin e/m	5.15
*CTSV*	Cathepsin V	6.69
*CYLD*	Cyld lysine 63 deubiquitinase	2.65
*DLX3*	Distal-less homeobox 3	2.08
*DSG1*	Desmoglein 1	15.85
*FLG*	Filaggrin	2.77
*HBP1*	HMG-box transcription factor 1	8.34
*IVL*	Involucrin	24.62
*KLF4*	Kruppel-like factor 4	3.90
*KLK5*	Kallikrein-related peptidase 5	22.74
*KLK7*	Kallikrein-related peptidase 7	39.73
*KRT1*	Keratin 1	2.37
*KRT13*	Keratin 13	78.42
*NCOA3*	Nuclear receptor coactivator 3	3.28
*PPL*	Periplakin	18.86
*SPINK5*	Serine protease inhibitor kazal type-5	6.77
*TGM1*	Transglutaminase 1	5.14
*TMEM79*	Transmembrane protein 79	3.11

In this category, 88 genes are listed as the altered genes. Based on the literature information, we selected 21 genes that were related to skin aging, keratinocyte differentiation and calcium signaling.

**Table 3 nutrients-13-02144-t003:** Major gene lists of “Calcium signal”.

Gene Symbol	Gene Name	Fold Change (eESM vs. Control)
*Cacna1a*	Calcium voltage-gated channel subunit alpha1 A	1.87
*Cacna1d*	Calcium voltage-gated channel subunit alpha1 D	1.59
*Cacna1g*	Calcium voltage-gated channel subunit alpha1 G	1.66
*Camk1d*	Calcium calmodulin-dependent protein kinase ID	1.75
*Camk2a*	Calcium calmodulin-dependent protein kinase II alpha	1.61
*Casq1*	calsequestrin 1	5.29
*Creb3l4*	cAMP responsive element binding protein 3-like 4	2.10
*Micu1*	Mitochondrial calcium uptake 1	1.57
*Prkacb*	protein kinase cAMP-activated catalytic subunit beta	1.53
*Prkar1b*	Protein kinase cAMP-dependent type I regulatory subunit beta	1.55
*Prkar2b*	Protein kinase cAMP-dependent type II regulatory subunit beta	1.59
*Trpc3*	Transient receptor potential cation channel subfamily C member 3	1.84
*Trpv6*	Transient receptor potential cation channel subfamily V member 6	1.70

In this category, 35 genes were listed up as the altered genes. Based on the literature information, we selected 13 genes that were related to the calcium signal.

## Data Availability

The microarray data have been deposited in the GEO database (https://www.ncbi.nlm.nih.gov/geo/ (accessed on 22 June 2021)) under accession number GSE178597.

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
