# Peer review of "Effects of Eggshell Membrane on Keratinocyte Differentiation and Skin Aging In Vitro and In Vivo"

_nutrients, 2021, doi:10.3390/nu13072144_

Round 1

Reviewer 1 Report

The authors describe the beneficial effects of an egg byproduct on skin appearance during aging. They suggest that the eggshell membrane can be used as a type of nutraceutical. The study is interesting, in the scope of the journal and provides new data (including microarray analysis in different contexts) that can be used in other future studies.

I only have some minor comments to make that should be addressed before the manuscript is found acceptable for publication.

Specific minor points

  1. What is the relevance of the Il10-/- mouse model in terms of the normal aging process? The authors need to discuss this issue further. Il10-/- mice are model for inflammatory disease in intestine.
  2. Why did the author only used male Il10-/- mice?
  3. How where the enzymatically and alkaline hydrolyzed ESM prepared? The authors should provide a small description on this.
  4. The study describes a process of “repositioning” in the extended version of this term as described very recently (Sotiropoulou G, Zingkou E, Pampalakis G (2021) Redirecting drug repositioning to discover innovative cosmeceutical. Exp Dermatol 30: 628-644) where compounds other than drugs are repositioned against diseases or cosmetics. This Reference should be added in reference list and discussed in the discussion part.
  5. Please define a, b, bc, c in Figures in terms of p-values.
  6. Kallikrein-related peptidases should be in capital letters for the human (KLK5 and KLK7).
  7. It would have been nice to check the expression of Klk5 and Klk7 in the epidermis of mice after application of ESM as they have done with keratinocyte cultures due to their major involvement in differentiation and normal physiology.

Reviewer 2 Report

In the present manuscript, Furkawa et al. investigate the nutritional effects of eggshell membrane and products thereof on human epidermal keratinocytes in vitro and mouse skin in vivo with special focus on cell and tissue aging.

Although most people in industrial, developed countries do not suffer from malnutrition, food supplements gain increasing popularity for cosmetic and anti-aging purposes. Thus, the present study is interesting with this respect. There are many open question concerning the way how additional food supplements benefit skin health compared to controls receiving complete nutrition. However, this problem is not addressed here.
The authors investigate direct effects of eggshell membrane products on one cell type (NHEK) in vitro and use one mouse strain with complete knockout of IL10 for in vivo studies. The choice of these models was not founded; dermal fibroblasts treated with senescence-associated secretory proteins (SASP) in vitro or other aging models in vivo would have been also of interest.

All methods are valid and were described precisely. Why did the authors use powder, hydrolysed and enzymatically digested ESM?

The use of Gapdh as a reference gene in nutritional studies seems to require prove that the treatment does not impair the expression of the reference gene as well. Can this prerequisite be demonstrated from the microarray dataset?

Statistics: The authors present their data as mean ± SEM from 4 independent experiments. However, data should be presented as mean value ±SD. SD shows the real deviation of the measurements, whereas SEM is not a statistical measure but depicts how precise the mean represents the group. SEM depends on the number of samples and does not reflect the biological variance. (Jaykaran, 2010, DOI: 10.4103/0253-7613.70402). Thus, all related diagrams have to be re-calculated and re-made with the right statistics.

Results:
The in vitro effects of eESM on NHEK were likely to be mediated by the PKC signalling pathway. This could have been shown easily by incubation NHK with eESM and a PKC-Inhibitor in parallel and would substantiate the dataset.

Fig.3: The skin histology pictures are rather small and the skin compartiments cannot be inspected: How thick are the epidermis and dermis separately? Where is the panniculus carnosus muscle? What are typical signs of skin aging in these IL10-KO-mice?
Which properties were improved by pESM treatment in mouse skin?

The array experiments show interesting results that can be discussed in the context of the in vitro data and selected published findings from aging skin. With this respect, the present study is sound and adds interesting facts supporting the functionality of ESM in mammal skin physiology. Unfortunately, the study lacks any mechanistic explanations of the effects observed. Due to restricted models used here, the universal validity of these findings remains to be proven.

Reviewer 3 Report

Manuscript No. 1240674 deals with the effect of the eggshell membrane on keratinocyte differentiation and skin aging in Vitro and in Vivo. This study showed that oral pESM supplementation is a source of many bioactive compounds that may have a beneficial effect on the skin. The authors showed that eESM in particular (enzymatically hydrolyzed form) can be proposed as a functional application of ESM in our life and health (e.g. skin cream).

General information

 - a well-written manuscript, understandable for the reader

- methodology and research results well described.

Reviewer's suggestions

- Figure 1 a. In the graph, the y axis should be described as cell cytotoxicity

- Fig. 1b - the posted photos are hardly legible (too low magnification makes it impossible to observe significant changes in cell morphology)

-Figure 3b, the y axis should be labeled as Skin thickness (µm)

-What does the abbreviation A.U stand for on the y-axis in d, e, f graphs (Figures 2 and 4)??

In my opinion, the methodology should include a description of the IPA system that was used in this study (bioinformatics analysis, which included canonical pathway analysis, disease and function, regulator effects, upstream regulators and molecular networks).
